# VarGrad: A Low-Variance Gradient Estimator for Variational Inference

**Lorenz Richter**\*
Freie Universität Berlin
BTU Cottbus-Senftenberg
dida Datenschmiede GmbH
lorenz.richter@fu-berlin.de

**Ayman Boustati**\*
University of Warwick
a.boustati@warwick.ac.uk

**Nikolas Nüsken**
Universität Potsdam
nuesken@uni-potsdam.de

**Francisco J. R. Ruiz**
DeepMind
franrruiz@google.com

**Ömer Deniz Akyildiz**
University of Warwick
The Alan Turing Institute
omer.akyildiz@warwick.ac.uk

## Abstract

We analyse the properties of an unbiased gradient estimator of the evidence lower bound (ELBO) for variational inference, based on the score function method with leave-one-out control variates. We show that this gradient estimator can be obtained using a new loss, defined as the variance of the log-ratio between the exact posterior and the variational approximation, which we call the *log-variance loss*. Under certain conditions, the gradient of the log-variance loss equals the gradient of the (negative) ELBO. We show theoretically that this gradient estimator, which we call *VarGrad* due to its connection to the log-variance loss, exhibits lower variance than the score function method in certain settings, and that the leave-one-out control variate coefficients are close to the optimal ones. We empirically demonstrate that VarGrad offers a favourable variance versus computation trade-off compared to other state-of-the-art estimators on a discrete variational autoencoder (VAE).

## 1 Introduction

Estimating the gradient of the expectation of a function is a problem with applications in many areas of machine learning, ranging from variational inference to reinforcement learning [Mohamed et al., 2019]. Different gradient estimators lead to different algorithms; two examples of estimators are the score function gradient [Williams, 1992] and the reparameterisation gradient [Kingma and Welling, 2014, Rezende et al., 2014, Titsias and Lázaro-Gredilla, 2014]. Many recent works develop new estimators with different properties (such as their variance); see Section 5 for a review.

We focus on variational inference (VI), where the goal is to approximate the posterior distribution $p(z \mid x)$ of a model $p(x, z)$, where $x$ denotes the observations and $z$ refers to the latent variables of the model [Jordan et al., 1999, Blei et al., 2017]. VI approximates the posterior using a parameterised family of distributions $q_\phi(z)$, and finds the parameters $\phi$ by minimising the Kullback-Leibler (KL)

---

divergence from $q_\phi(z)$ to $p(z \mid x)$, i.e., $\mathrm{KL}(q_\phi(z) \mid\mid p(z \mid x))$. Since the KL is intractable, VI solves instead an equivalent problem that maximises the evidence lower bound (ELBO), given by

$$\mathrm{ELBO}(\phi) = \mathbb{E}_{q_\phi}\left[\log \frac{p(x,z)}{q_\phi(z)}\right]. \tag{1}$$

Thus, VI casts the inference problem as an optimisation problem, which can be solved with stochastic optimisation tools when the ELBO is not available in closed form. In particular, VI forms a Monte Carlo estimator of the gradient of the ELBO, $\nabla_\phi \mathrm{ELBO}(\phi)$.

In this paper, we analyse a multi-sample estimator of the gradient of the ELBO. In particular, we focus on an estimator first introduced by Salimans and Knowles [2014] and Kool et al. [2019], which is based on the score function method [Williams, 1992] with leave-one-out control variates.

We first show the connection between this estimator and an alternative divergence measure between the variational distribution $q_\phi(z)$ and the exact posterior $p(z \mid x)$. This divergence, which is different from the standard KL used in variational inference, is defined as the variance, under some arbitrary distribution $r(z)$, of the log-ratio $\log \frac{q_\phi(z)}{p(z \mid x)}$. We refer to this divergence as the *log-variance loss*. Inspired by Nüsken and Richter [2020], we show that we recover the gradient estimator of Salimans and Knowles [2014] and Kool et al. [2019] by taking the gradient with respect to the variational parameters $\phi$ of the log-variance loss and evaluating the result at $r(z) = q_\phi(z)$. Due to this property, we refer to the gradient estimator as *VarGrad*. This property also suggests a simple algorithm for computing the gradient estimator, based on differentiating through the log-variance loss.

We then study the relationship between VarGrad and the score function estimator [Williams, 1992, Carbonetto et al., 2009, Paisley et al., 2012, Ranganath et al., 2014] with optimal control variate coefficients. We show that the control variate coefficients of VarGrad are close to the (intractable) optimal coefficients. Indeed, we show both theoretically and empirically that the difference between both is small in many cases; for example when the KL from $q_\phi(z)$ to the posterior is either small or large, which is generally the case in the late and early stages of the optimisation, respectively. This explains the success of the VarGrad estimator in a variety of settings [Kool et al., 2019, 2020].

Since it is based on the score function, VarGrad is a black-box, general purpose estimator because it makes no assumptions on the model $p(x, z)$, such as differentiability with respect to the latent variables $z$. It introduces no additional parameters to be tuned and it is not computationally expensive. In Section 6, we show empirically that VarGrad exhibits a favourable variance versus computation trade-off compared to other unbiased gradient estimators, including the score function gradient with control variates [Williams, 1992, Ranganath et al., 2014], REBAR [Tucker et al., 2017], RELAX [Grathwohl et al., 2018], and ARM [Yin and Zhou, 2019].

## 2 Background

In this section, we introduce the notation and review one of the most relevant estimators in VI: the score function method. We also review its improved version based on leave-one-out control variates.

Consider a probabilistic model $p(x, z)$, where $z$ denotes the latent variables and $x$ are the observations. We are interested in computing the posterior $p(z \mid x) = p(x, z)/p(x)$, where $p(x) = \int p(x, z)\, \mathrm{d}z$ is the marginal likelihood. For most models of interest, the posterior is intractable due to the intractability of the marginal likelihood, and we resort to an approximation.

Variational inference approximates the posterior $p(z \mid x)$ with a parameterised family of distributions $q_\phi(z)$ (with $\phi \in \Phi$), called variational family. Variational inference finds the parameters $\phi^*$ that minimise the KL divergence, $\phi^* = \operatorname{argmin}_{\phi \in \Phi} \mathrm{KL}\left(q_\phi(z) \mid\mid p(z \mid x)\right)$. This optimisation problem is intractable because the KL itself depends on the intractable posterior. Variational inference sidesteps this problem by maximising instead the ELBO defined in Eq. 1, which is a lower bound on the marginal likelihood, since $\log p(x) = \mathrm{ELBO}(\phi) + \mathrm{KL}\left(q_\phi(z) \mid\mid p(z \mid x)\right)$. As the expectation in Eq. 1 is typically intractable, variational inference uses stochastic optimisation to maximise the ELBO. In particular, it forms unbiased Monte Carlo estimators of the gradient $\nabla_\phi \mathrm{ELBO}(\phi)$.

We next review the score function method, a Monte Carlo estimator commonly used in variational inference. Instead of the ELBO, we focus on the gradients of the KL divergence with respect to the variational parameters $\phi$. These gradients are equal to the gradients of the negative ELBO because the marginal likelihood $p(x)$ does not depend on $\phi$; that is, $\nabla_\phi \mathrm{KL}\left(q_\phi(z) \mid\mid p(z \mid x)\right) = -\nabla_\phi \mathrm{ELBO}(\phi)$.

The score function estimator [Williams, 1992, Carbonetto et al., 2009, Paisley et al., 2012, Ranganath et al., 2014], also known as Reinforce, expresses the gradient as an expectation that depends on the log-ratio $q_\phi(z)/p(x,z)$ weighted by the score function $\nabla_\phi \log q_\phi(z)$. The resulting estimator is

$$\nabla_\phi \text{KL}\left[q_\phi(z) \,||\, p(z\,|\,x)\right] \approx \widehat{g}_{\text{Reinforce}}(\phi) = \frac{1}{S}\sum_{s=1}^{S}\log\left(\frac{q_\phi(z^{(s)})}{p(x,z^{(s)})}\right)\nabla_\phi \log q_\phi(z^{(s)}), \qquad (2)$$

where $z^{(s)} \overset{\text{i.i.d.}}{\sim} q_\phi(z)$. Due to its high variance, the score function estimator requires additional tricks in practice; Ranganath et al. [2014] use Rao-Blackwellization and control variates. The control variates are multiples of the score function, $a \odot \nabla_\phi \log q_\phi(z)$, where $\odot$ denotes the Hadamard (element-wise) product and the coefficient $a$ is chosen to minimise the estimator variance.

Salimans and Knowles [2014] and Kool et al. [2019] leverage the multi-sample estimator by using $S-1$ samples to compute the control variate coefficient $a$ and then average over the resulting estimators, obtaining a leave-one-out estimator

$$\widehat{g}_{\text{LOO}}(\phi) = \frac{1}{S-1}\left(\sum_{s=1}^{S}f_\phi(z^{(s)})\nabla_\phi \log q_\phi(z^{(s)}) - \bar{f}_\phi \sum_{s=1}^{S}\nabla_\phi \log q_\phi(z^{(s)})\right), \quad z^{(s)} \overset{\text{i.i.d.}}{\sim} q_\phi(z), \quad (3)$$

where for simplicity of notation we have defined $f_\phi(z)$ as the log-ratio $\log \frac{q_\phi(z)}{p(x,z)}$ and $\bar{f}_\phi$ as its empirical average (which is a multi-sample Monte Carlo estimate of the negative ELBO), i.e.,

$$f_\phi(z) = \log \frac{q_\phi(z)}{p(x,z)} \quad \text{and} \quad \bar{f}_\phi = \frac{1}{S}\sum_{s=1}^{S}f_\phi(z^{(s)}) \approx -\text{ELBO}(\phi). \qquad (4)$$

The score function method makes no assumptions on the model $p(x,z)$ or the distribution $q_\phi(z)$; the only requirements are to be able to sample from $q_\phi(z)$ and to evaluate $\log q_\phi(z)$ and $\log p(x,z)$.

## 3 The Log-Variance Loss and its Connection to VarGrad

In this section, we show the connection between the leave-one-out estimator in Eq. 3 and a novel divergence, which we call the log-variance loss. We introduce the log-variance loss in Section 3.1 and show its connection to Eq. 3 in Section 3.2. We refer to the estimator in Eq. 3 as *VarGrad*.

### 3.1 The Log-Variance Loss

The log-variance loss is defined as the variance, under some arbitrary distribution $r(z)$, of the log-ratio $\log \frac{q_\phi(z)}{p(z\,|\,x)}$. It has the property of reproducing the gradients of the KL divergence under certain conditions (see Proposition 1 for details). We next give the precise definition of the loss.

**Definition 1.** *For a given distribution $r(z)$, the log-variance loss $\mathcal{L}_r(\cdot)$ is given by*

$$\mathcal{L}_r(q_\phi(z) \,||\, p(z\,|\,x)) = \frac{1}{2}\text{Var}_r\left(\log\left(\frac{q_\phi(z)}{p(z\,|\,x)}\right)\right). \qquad (5)$$

We refer to the distribution $r(z)$ as the *reference distribution* under which the discrepancy between $q_\phi(z)$ and the posterior $p(z\,|\,x)$ is computed. When the support of the reference distribution contains the supports of $q_\phi(z)$ and $p(z\,|\,x)$, Eq. 5 is a divergence;[†] it is zero if and only if $q_\phi(z) = p(z\,|\,x)$. The factor $1/2$ in Eq. 5 is only included because it simplifies some expressions later in this section.

We next show that the gradient of the log-variance loss and the gradient of the standard KL divergence coincide under certain conditions. In particular, taking the gradient of Eq. 5 with respect to the variational parameters $\phi$ *and then* evaluating the result for a reference distribution $r(z) = q_\phi(z)$ gives the gradient of the KL. This property is detailed in Proposition 1.

---

[†]More technically, as we assume that $r(z)$, $p(z\,|\,x)$, and $q_\phi(z)$ admit densities, it follows that measure-zero sets of $r(z)$ are necessarily measure-zero sets of $p(z\,|\,x)$ and $q_\phi(z)$, implying that the divergence is well defined.

**Proposition 1.** *The gradient with respect to $\phi$ of the log-variance loss, evaluated at $r(z) = q_\phi(z)$, equals the gradient of the* KL *divergence,*

$$\nabla_\phi \mathcal{L}_r(q_\phi(z) \mid\mid p(z \mid x))\Big|_{r=q_\phi} = \nabla_\phi \mathrm{KL}(q_\phi(z) \mid\mid p(z \mid x)). \tag{6}$$

*Proof.* See Appendix A.1. □

Proposition 1 implies that we can estimate the gradient of the KL divergence by estimating instead the gradient of the log-variance loss.

**Remark 1.** The result in Proposition 1 is obtained by setting $r(z) = q_\phi(z)$ *after* taking the gradient with respect to $\phi$. The same result does not hold if we set $r(z) = q_\phi(z)$ before differentiating.

### 3.2 VarGrad: Derivation of the Gradient Estimator from the Log-Variance Loss

The leave-one-out estimator in Eq. 3 [Salimans and Knowles, 2014, Kool et al., 2019] is connected to the log-variance loss from Section 3.1 through Proposition 1. Firstly, note that the log-variance loss is intractable as it depends on the posterior $p(z \mid x)$. However, since the marginal likelihood $p(x)$ has zero variance, it can be dropped from the definition in Eq. 5, yielding

$$\mathcal{L}_r(q_\phi(z) \mid\mid p(z \mid x)) = \frac{1}{2} \mathrm{Var}_r \left( \log \left( \frac{q_\phi(z)}{p(x, z)} \right) \right) = \frac{1}{2} \mathrm{Var}_r \left( f_\phi(z) \right), \tag{7}$$

where $f_\phi(z)$ is defined in Eq. 4.

Next, we build the estimator of the log-variance loss as the empirical variance of $S$ Monte Carlo samples,

$$\mathcal{L}_r(q_\phi(z) \mid\mid p(z \mid x)) \approx \frac{1}{2(S-1)} \sum_{s=1}^{S} \left( f_\phi(z^{(s)}) - \bar{f}_\phi \right)^2, \quad z^{(s)} \overset{\text{i.i.d.}}{\sim} r(z). \tag{8}$$

Applying Proposition 1 by differentiating through Eq. 8, we arrive at the VarGrad estimator, $\widehat{g}_{\mathrm{VarGrad}}(\phi) = \widehat{g}_{\mathrm{LOO}}(\phi) \approx \nabla_\phi \mathrm{KL}\left(q_\phi(z) \mid\mid p(z \mid x)\right)$, where

$$\widehat{g}_{\mathrm{VarGrad}}(\phi) = \frac{1}{S-1} \left( \sum_{s=1}^{S} f_\phi(z^{(s)}) \nabla_\phi \log q_\phi(z^{(s)}) - \bar{f}_\phi \sum_{s=1}^{S} \nabla_\phi \log q_\phi(z^{(s)}) \right) \tag{9}$$

and $z^{(s)} \overset{\text{i.i.d.}}{\sim} q_\phi(z)$.

The expression for VarGrad in Eq. 9 is identical to that of the leave-one-out estimator in Eq. 3. Thus, VarGrad is an unbiased estimator of the gradient of the KL (and equivalently the gradient of the ELBO). From a probabilistic programming perspective, setting the reference $r(z) = q_\phi(z)$ *after* differentiating w.r.t. $\phi$ amounts to sampling $z^{(s)} \sim q_\phi(z)$ and detaching the resulting samples from the computational graph. This suggests a novel algorithmic procedure, given in Algorithm 1. Its implementation is simple: we only need the samples $z^{(s)} \sim q_\phi(z)$ and apply the `stop_gradient` operator, evaluate the log-ratio $f_\phi(z^{(s)})$ for each sample, and then differentiate through the empirical variance of this log-ratio.

---

**Algorithm 1** Pseudocode for VarGrad

---

**Input:** Variational parameters $\phi$, data $x$
    **for** $s = 1, \ldots, S$ **do**
        $z^{(s)} \leftarrow \texttt{sample}(q_\phi(\cdot))$                ▷ Sample from the approximate posterior
        $z^{(s)} \leftarrow \texttt{stop\_gradient}(z^{(s)})$      ▷ Detach the samples from the computational graph
        $f_\phi^{(s)} \leftarrow \log q_\phi(z^{(s)}) - \log p(x, z^{(s)})$      ▷ An estimate of the negative ELBO
    $\widehat{\mathcal{L}} \leftarrow \frac{1}{2} \texttt{Variance}(\{f_\phi^{(s)}\}_{s=1}^{S})$          ▷ An estimate of the log-variance loss
    **return** $\texttt{grad}(\widehat{\mathcal{L}})$               ▷ Differentiate through the loss w.r.t. $\phi$

---

# 4  Analytical Results

In this section we study the properties of $\widehat{g}_{\text{VarGrad}}$ in comparison to other estimators based on the score function method. In Section 4.1, we analyse the difference $\delta^{\text{CV}}$ between the control variate coefficient of VarGrad (called $a^{\text{VarGrad}}$) and the optimal one. The former can be approximated cheaply and unbiasedly, while a standard Monte Carlo estimator of the latter is biased and often exhibits high variance. Furthermore, we establish that the difference $\delta^{\text{CV}}$ is negligible in certain settings, in particular when $\text{KL}(q_\phi(z) \,||\, p(z\,|\,x))$ is either very large or close to zero; thus in these settings the control variate coefficient of VarGrad is close to the optimal coefficient. In Section 4.2 we show that a simple relation between $\delta^{\text{CV}}$ and the ELBO is sufficient to *guarantee* that $\widehat{g}_{\text{VarGrad}}$ has lower variance than $\widehat{g}_{\text{Reinforce}}$ when the number of Monte Carlo samples is large enough.

## 4.1  Analysis of the Control Variate Coefficients

As alluded to in Section 2, Ranganath et al. [2014] proposed to modify $\widehat{g}_{\text{Reinforce}}$ using a score function control variate, that is,

$$\widehat{g}_{\text{CV}}(\phi) = \widehat{g}_{\text{Reinforce}}(\phi) - a \odot \left( \frac{1}{S} \sum_{s=1}^{S} \nabla_\phi \log q_\phi(z^{(s)}) \right), \tag{10}$$

where $a$ is a vector chosen so as to reduce the variance of the estimator. We recover VarGrad (Eq. 9), up to a factor of proportionality, by setting the control variate coefficient $a = \bar{f}_\phi \mathbf{1}$ in Eq. 10, where $\mathbf{1}$ is a vector of ones. The proportionality relation is $\frac{S-1}{S}\widehat{g}_{\text{CV}} = \widehat{g}_{\text{VarGrad}}$. In terms of variance reduction, the coefficients of the optimal $a^*$ are given by

$$a_i^* = \frac{\text{Cov}_{q_\phi}\left(f_\phi \partial_{\phi_i} \log q_\phi, \partial_{\phi_i} \log q_\phi\right)}{\text{Var}_{q_\phi}\left(\partial_{\phi_i} \log q_\phi\right)}. \tag{11}$$

We next show that the coefficients of VarGrad, $a^{\text{VarGrad}}$, are close to the optimal coefficients $a^*$. For this, we first relate $a^{\text{VarGrad}}$ to $a^*$ in Lemma 1.

**Lemma 1.** *We can write the optimal control variate coefficient as the expected value of $a^{VarGrad}$ plus a* control variate correction *term $\delta^{CV}$, i.e.,*

$$a^* = \mathbb{E}_{q_\phi}[a^{VarGrad}] + \delta^{CV} = -\text{ELBO}(\phi) + \delta^{CV}, \tag{12}$$

*where $a^{VarGrad} = \bar{f}_\phi$ and the components of the correction term $\delta^{CV}$ are given by*

$$\delta_i^{CV} = \frac{\text{Cov}_{q_\phi}\left(f_\phi, (\partial_{\phi_i} \log q_\phi)^2\right)}{\text{Var}_{q_\phi}\left(\partial_{\phi_i} \log q_\phi\right)}. \tag{13}$$

*Proof.* See Appendix A.2. □

According to Lemma 1, the difference between the optimal control variate coefficient and the (expected) VarGrad coefficient is equal to the correction $\delta^{\text{CV}}$. We hypothesise that direct Monte Carlo estimation of $\delta^{\text{CV}}$ in Eq. 13 (or similarly for Eq. 11) suffers from high variance because it takes the form of a fraction[‡] (see for instance Appendix C). Moreover, estimating Eq. 13 by taking the ratio of two Monte Carlo estimators gives a biased estimate.

We next show that in certain settings the correction term $\delta^{\text{CV}}$ becomes negligible, implying that $\widehat{g}_{\text{VarGrad}}$ and $\widehat{g}_{\text{Reinforce}}$ equipped with the optimal control variate coefficients behave almost identically. We provide empirical evidence of this finding in Section 6 (and in Appendix C for the Gaussian case).

**Proposition 2** ($\delta^{\text{CV}}$ is small in comparison to $\mathbb{E}_{q_\phi}[a^{\text{VarGrad}}]$ if the KL divergence between $q_\phi(z)$ and $p(z\,|\,x)$ is large or small)**.** *Assume that $q_\phi(z)$ has lighter tails than the posterior $p(z\,|\,x)$, in the sense that there exists a constant $C > 0$ such that*

$$\sup_z \frac{q_\phi(z)}{p(z\,|\,x)} < C. \tag{14}$$

---

[‡]Monte Carlo estimators of fractions are not straightforward. As a simple example, consider the ratio of two independent Gaussian random variables, each with zero mean and unit variance. The ratio follows a Cauchy distribution, which has infinite variance.

*Furthermore, define the kurtosis of the score function,*

$$\mathrm{Kurt}[\partial_{\phi_i} \log q_\phi] = \frac{\mathbb{E}_{q_\phi}[(\partial_{\phi_i} \log q_\phi)^4]}{(\mathbb{E}_{q_\phi}[(\partial_{\phi_i} \log q_\phi)^2])^2}, \tag{15}$$

*and assume that it is bounded,* $\mathrm{Kurt}[\partial_{\phi_i} \log q_\phi] < \infty$. *Then, the ratio between the control variate correction* $\delta^{CV}$ *and the expected control variate coefficient of VarGrad can be upper bounded by*

$$\left| \frac{\delta_i^{CV}}{\mathbb{E}_{q_\phi}[a^{VarGrad}]} \right| \leq \frac{2\sqrt{C\,\mathrm{Kurt}[\partial_{\phi_i} \log q_\phi]}}{\left| \sqrt{\mathrm{KL}(q_\phi(z) \,||\, p(z\,|\,x))} - \frac{\log p(x)}{\sqrt{\mathrm{KL}(q_\phi(z) \,||\, p(z\,|\,x))}} \right|}. \tag{16}$$

*Proof.* See Appendix A.3. $\square$

**Remark 2.** The variational approximation $q_\phi(z)$ typically underestimates the spread of the posterior $p(z\,|\,x)$ [Blei et al., 2017], and so the assumption in Eq. 14 is typically satisfied in practice after a few iterations of the optimisation algorithm. The kurtosis $\mathrm{Kurt}[\partial_{\phi_i} \log q_\phi]$ quantifies the weight of the tails of the variational approximation in terms of the score function. In Appendix A.6 we analyse the kurtosis of exponential family distributions and show that it is uniformly bounded for Gaussian variational families.

**Remark 3.** The upper bound in Eq. 16 allows us to identify two regimes. When $\mathrm{KL}(q_\phi(z) \,||\, p(z\,|\,x))$ is large, the bound asserts that the relative error satisfies

$$\left| \frac{\delta_i^{\mathrm{CV}}}{\mathbb{E}_{q_\phi}[a^{\mathrm{VarGrad}}]} \right| \lessapprox \mathcal{O}\left( \mathrm{KL}(q_\phi(z) \,||\, p(z\,|\,x))^{-1/2} \right), \tag{17}$$

as the second term in the denominator of Eq. 16 becomes negligible. This can happen in the early stages of the optimisation process, in which case we can conclude that $\delta^{\mathrm{CV}}$ is expected to be small. Since the KL divergence increases with the dimensionality of the latent variable $z$ (see Appendix A.7), Eq. 16 also implies that the ratio becomes smaller as the number of latent variables grows. Moreover, if the *minimum* KL divergence between the variational family and the true posterior is large (i.e., if the best candidate in the variational family is still far away from the target), the correction term $\delta_i^{\mathrm{CV}}$ can be negligible during the whole optimisation procedure, which is often the case in practice.

In the regime where $\mathrm{KL}(q_\phi(z) \,||\, p(z\,|\,x))$ approaches zero (i.e., towards the end of the optimisation process if the variational family is well specified and includes the posterior), then Eq. 16 implies that

$$\left| \frac{\delta_i^{\mathrm{CV}}}{\mathbb{E}_{q_\phi}[a^{\mathrm{VarGrad}}]} \right| \lessapprox \mathcal{O}\left( \mathrm{KL}(q_\phi(z) \,||\, p(z\,|\,x))^{1/2} \right). \tag{18}$$

In this regime, the error w.r.t. the optimal control variate coefficient decreases with the KL divergence. The estimates in Eq. 17 and Eq. 18 combined suggest that the relative error remains bounded throughout the optimisation. We verify this proposition experimentally in Section 6.

## 4.2 Variance of the Estimator

In this section we provide a result that guarantees that the variance of $\widehat{g}_{\mathrm{VarGrad}}$ is smaller than the variance of $\widehat{g}_{\mathrm{Reinforce}}$ when the number of Monte Carlo samples is large enough.

**Proposition 3.** *Consider the two gradient estimators* $\widehat{g}_{Reinforce}(\phi)$ *and* $\widehat{g}_{VarGrad}(\phi)$, *each with S Monte Carlo samples, as defined in Eq. 2 and Eq. 9, respectively. If*

$$-\frac{\delta_i^{CV}}{\mathbb{E}_{q_\phi}[a^{VarGrad}]} = \frac{\delta_i^{CV}}{\mathrm{ELBO}(\phi)} < \frac{1}{2} \tag{19}$$

*then there exists* $S_0 \in \mathbb{N}$ *such that*

$$\mathrm{Var}\left( \widehat{g}_{VarGrad,i}(\phi) \right) \leq \mathrm{Var}\left( \widehat{g}_{Reinforce,i}(\phi) \right), \qquad \text{for all} \quad S \geq S_0. \tag{20}$$

*Proof.* See Appendix A.4. $\square$

If the correction $\delta^{\text{CV}}$ is negligible in the sense of Proposition 2, then the assumption in Eq. 19 is satisfied and Proposition 3 guarantees that VarGrad has lower variance than Reinforce when $S$ is large enough. We arrive at the following corollary, which also considers the dimensionality of the latent variables. The main assumption -that the KL-divergence increases with the dimension of the latent space- is supported by the result in Appendix A.7.

**Corollary 1.** *Let $S$ be the number of samples and $D$ the dimension of the latent variable $z$. Furthermore, let the assumptions of Proposition 2 be satisfied and assume that $\text{KL}(q_\phi(z) \,||\, p(z \,|\, x))$ is strictly increasing in $D$. Then, there exist $S_0, D_0 \in \mathbb{N}$ such that*

$$\text{Var}\left(\widehat{g}_{\text{VarGrad},i}(\phi)\right) \leq \text{Var}\left(\widehat{g}_{\text{Reinforce},i}(\phi)\right), \qquad \text{for all } S \geq S_0 \text{ and } D \geq D_0. \tag{21}$$

*Proof.* See Appendix A.5. □

We provide further intuition on the condition in Eq. 19 with the analysis in Appendix C.1.

## 5   Related Work

In the last few years, many gradient estimators of the ELBO have been proposed; see Mohamed et al. [2019] for a comprehensive review. Among those, the score function estimators [Williams, 1992, Carbonetto et al., 2009, Paisley et al., 2012, Ranganath et al., 2014] and the reparameterisation estimators [Kingma and Welling, 2014, Rezende et al., 2014, Titsias and Lázaro-Gredilla, 2014], as well as combinations of both [Ruiz et al., 2016, Naesseth et al., 2017], are arguably the most widely used. NVIL [Mnih and Gregor, 2014] and MuProp [Gu et al., 2016] are unbiased gradient estimators for training stochastic neural networks.

Other gradient estimators are specific for discrete-valued latent variables. The concrete relaxation [Maddison et al., 2017, Jang et al., 2017] described a way to form a biased estimator of the gradient, which REBAR [Tucker et al., 2017] and RELAX [Grathwohl et al., 2018] use as a control variate to obtain an unbiased estimator. Other recent estimators have been proposed by Lee et al. [2018], Peters and Welling [2018], Shayer et al. [2018], Cong et al. [2019], Yin and Zhou [2019], Yin et al. [2019], and Dong et al. [2020]. In Section 6, we compare VarGrad with some of these estimators, showing that it exhibits a favourable performance versus computational complexity trade-off.

The VarGrad estimator was first introduced by Salimans and Knowles [2014] and Kool et al. [2019]. It also relates to VIMCO [Mnih and Rezende, 2016] in that it is a leave-one-out estimator. In this paper, we have described an alternative derivation of VarGrad, based on the log-variance loss.

The log-variance loss from Section 3.1 defines an alternative divergence between the approximate and the exact posterior distributions. In the context of optimal control of diffusion processes and related forward-backward stochastic differential equations, it arises naturally to quantify the discrepancy between measures on path space [Nüsken and Richter, 2020]. Other forms of alternative divergences have also been explored in previous work; for example the $\chi^2$-divergence [Dieng et al., 2017], the Rényi divergence [Li and Turner, 2016], the Langevin-Stein [Ranganath et al., 2016], the $\alpha$-divergence [Hernández-Lobato et al., 2016], other $f$-divergences [Wang et al., 2018], a contrastive divergence [Ruiz and Titsias, 2019], and also the inclusive KL [Naesseth et al., 2020], see also Appendix D.

Finally, from an implementation perspective, Algorithm 1 contains a `stop_gradient` operator that resembles the method of Roeder et al. [2017]. In Roeder et al. [2017], this operator is used on the variational parameters to eliminate the entropy term in the reparameterisation gradient to reduce its variance; this has also been extended to importance weighted variational objectives [Tucker et al., 2018]. In contrast, VarGrad applies the `stop_gradient` operator on the samples and is based on the score function method.

## 6   Experiments

In order to verify the properties of VarGrad empirically, we test it on two popular models: a Bayesian logistic regression model on a synthetic dataset and a discrete variational autoencoder (DVAE)

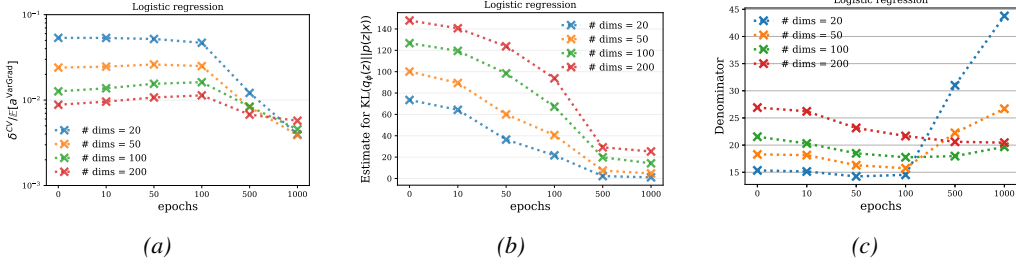

*(a)*                                 *(b)*                                 *(c)*

*Figure 1.* Illustration of Proposition 2 and Remark 3 on the logistic regression model. In (a), we show that the ratio $\left|\delta_i^{\text{CV}}/\mathbb{E}_{q_\phi}[a^{\text{VarGrad}}]\right|$ is small and uniformly bounded over epochs, illustrating that the VarGrad estimator stays close to the optimal control variate coefficients during the whole optimisation procedure. Additionally, this ratio decreases with increasing dimensionality of the latent variables. In (b), we display an estimate of the KL divergence across epochs and demonstrate the beneficial effect of higher dimensions, since the bound of Eq. 16 is expected to scale like $\mathcal{O}(\text{KL}^{-1/2})$ in the early phase. In (c), we plot an estimate of the denominator of the bound (Eq. 16), which increases or stays constant over epochs, demonstrating that the ratio in Eq. 16 stays stable (and small) over epochs.

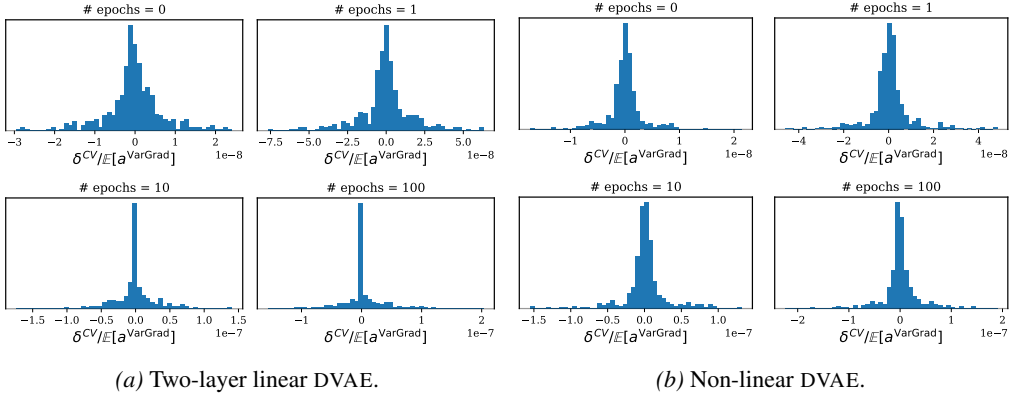

*(a)* Two-layer linear DVAE.                    *(b)* Non-linear DVAE.

*Figure 2.* The distribution of $\frac{\delta_i^{\text{CV}}}{\mathbb{E}[a^{\text{VarGrad}}]}$ associated with the biases of two DVAE models with 200 latent dimensions trained on Omniglot using VarGrad. The estimates are obtained with 2,000 Monte Carlo samples. The ratio $\frac{\delta_i^{\text{CV}}}{\mathbb{E}[a^{\text{VarGrad}}]}$ is consistently small throughout the optimisation procedure.

[Salakhutdinov and Murray, 2008, Kingma and Welling, 2014] on a fixed binarisation of Omniglot [Lake et al., 2015]. All details of the experiments can be found in Appendix B.[§]

**Closeness to the optimal control variate.** In Section 4 we analytically showed that VarGrad is close to the optimal control variate, and in particular that the ratio $\left|\delta_i^{\text{CV}}/\mathbb{E}_{q_\phi}[a^{\text{VarGrad}}]\right|$ can be small over the whole optimisation procedure. This behaviour is expected to be even more pronounced with growing dimensionality of the latent space. In Figure 1, we confirm this result by showing the ratio $\left|\delta_i^{\text{CV}}/\mathbb{E}_{q_\phi}[a^{\text{VarGrad}}]\right|$ for the logistic regression model. We also show the KL divergence along the iterations and the denominator of the bound in Eq. 16; see Figure 1 for the details.

In Figure 2, we provide further evidence that this ratio is also small when fitting DVAEs. Indeed, we observe that the ratio $\delta_i^{\text{CV}}/\mathbb{E}[a^{\text{VarGrad}}]$ is typically very small and is distributed around zero during the whole optimisation procedure.

**Variance reduction and computational cost.** In Figure 3 we show the variance of different gradient estimators throughout the optimisation in the logistic regression setting. We realise a significant improvement of VarGrad compared to the standard Reinforce estimator (Eq. 2). In fact, we observe a small difference between the variance of VarGrad and the variance of an *oracle estimator* based on Reinforce with access to the optimal control variate coefficient $a^*$. Figure 3 also shows the variance

---

[§]Code in JAX [Bradbury et al., 2018, Hennigan et al., 2020] is available at https://github.com/aboustati/vargrad.

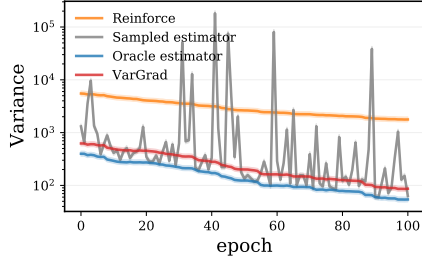

*Figure 3.* Estimates of the variance of the gradient component w.r.t. the posterior mean of one of the weights for the logistic regression model. The variance of VarGrad is close to the *oracle estimator* based on Reinforce with access to the optimal control variate coefficient $a^*$. Moreover, the *sampled estimator* (based on Reinforce with an estimate of $a^*$) shows the difficulty of estimating the optimal control variate coefficient in practice.

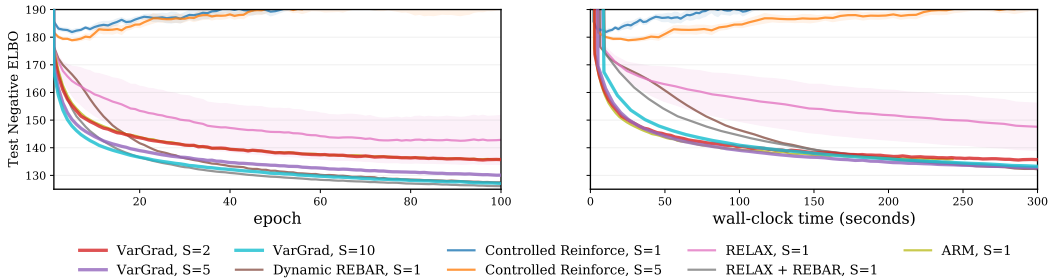

*Figure 4.* Optimisation trace versus epoch (left) and wall-clock time (right) for a two-layer linear DVAE on a fixed binarisation of Omniglot. The plot compares VarGrad to Reinforce with score function control variates [Ranganath et al., 2014], dynamic REBAR [Tucker et al., 2017], RELAX, RELAX + REBAR [Grathwohl et al., 2018] and ARM [Yin and Zhou, 2019]. The number of samples used to compute each gradient estimator is given in the figure legend. VarGrad demonstrates favourable scalability and performance when compared to the other estimators.

of the *sampled estimator*, which is based on Reinforce with an estimate of the optimal control variate; this confirms the difficulty of estimating it in practice. (A similar trend can be observed for the DVAE in the results in Appendix B, where VarGrad is compared to a wider list of estimators from the DVAE literature.) All methods use $S = 4$ Monte Carlo samples, and the control variate coefficient is estimated with either 2 extra samples (*sampled estimator*) or 1,000 samples (*oracle estimator*).

Finally, Figure 4 compares VarGad with other estimators by training a DVAE on Omniglot. The figure shows the negative ELBO as a function of the epoch number (left plot) and against the wall-clock time (right plot). The negative ELBO is computed on the standard test split and the optimisation uses Adam [Kingma and Ba, 2015] with learning rate of 0.001. VarGrad achieves similar performance to state-of-the-art estimators, such as REBAR [Tucker et al., 2017], RELAX [Grathwohl et al., 2018], and ARM [Yin and Zhou, 2019], while being simpler to implement (see Algorithm 1) and without any tunable hyperparameters.

## 7    Conclusions

We have analysed the VarGrad estimator, an estimator of the gradient of the KL that is based on Reinforce with leave-one-out control variates, which was first introduced by Salimans and Knowles [2014] and Kool et al. [2019]. We have established the connection between VarGrad and a novel divergence, which we call the log-variance loss. We have showed theoretically that, under certain conditions, the VarGrad control variate coefficients are close to the optimal ones. Moreover, we have established the conditions that guarantee that VarGrad exhibits lower variance than Reinforce. We leave it for future work to explore the direct optimisation of the log-variance loss for alternative choices of the reference distribution $r(z)$.

## Acknowledgments

Ö. D. A. and A. B. are funded by the Lloyds Register Foundation programme on Data Centric Engineering through the London Air Quality project at The Alan Turing Institute for Data Science and AI. This work was supported under EPSRC grant EP/N510129/1 as well as by Deutsche Forschungsgemeinschaft (DFG) through the grant CRC 1114 'Scaling Cascades in Complex Systems' (projects A02 and A05, project number 235221301).

## Broader Impact

Variational inference algorithms are approximate inference methods used in many practical applications, including computational neuroscience, natural language processing, and computer vision; see Blei et al. [2017]. The performance of variational inference often depends on the variance of the gradient estimator of its objective. High-variance estimators can make the resulting algorithm unstable and unreliable to use in real-world deployment scenarios; hence, deeper theoretical and empirical understanding of different gradient estimators is crucial for safe applicability of these methods in real-world settings.

In our work, we provide an analysis of the VarGrad estimator. We show the connection between this estimator and the "log-variance loss" and demonstrate its relationship to the optimal score function control variate for Reinforce, providing a theoretical analysis. We support our theoretical analysis with empirical results. We believe our work contributes in the further understanding of variational inference methods and contributes to improving their safety and applicability.

While our theoretical results are suggestive, we warn that the bounds depend on the nature of the model at hand. Therefore, the use of the results here while ignoring the model-specific aspects of our assumptions may lead to some risks in application.

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
