[Supplementary Material]

# Supplementary Material

## A Proof of Paper Results

### A.1 Proof of Proposition 1

*Proof.* We first consider the gradient of the KL divergence. It is given by

$$\nabla_\phi \text{KL}(q_\phi(z) \,||\, p(z\,|\,x)) = \int \nabla_\phi q_\phi(z)\,\mathrm{d}z + \int \log\left(\frac{q_\phi(z)}{p(z\,|\,x)}\right)\nabla_\phi q_\phi(z)\,\mathrm{d}z, \qquad (22)$$

where we can drop the first term since $\int \nabla_\phi q_\phi(z)\,\mathrm{d}z = \nabla_\phi \int q_\phi(z)\,\mathrm{d}z = \nabla_\phi(1) = 0$.

We now consider the gradient of the log-variance loss. Using the definition from Eq. 5, we see that

$$\nabla_\phi \mathcal{L}_r(q_\phi(z) \,||\, p(z\,|\,x)) = \frac{1}{2}\nabla_\phi \int \log^2\left(\frac{q_\phi(z)}{p(z\,|\,x)}\right)r(z)\,\mathrm{d}z - \frac{1}{2}\nabla_\phi\left(\int \log\left(\frac{q_\phi(z)}{p(z\,|\,x)}\right)r(z)\,\mathrm{d}z\right)^2$$

$$= \int \log\left(\frac{q_\phi(z)}{p(z\,|\,x)}\right)\frac{\nabla_\phi q_\phi(z)}{q_\phi(z)}r(z)\,\mathrm{d}z - \left(\int \log\left(\frac{q_\phi(z)}{p(z\,|\,x)}\right)r(z)\,\mathrm{d}z\right)\left(\int \frac{\nabla_\phi q_\phi(z)}{q_\phi(z)}r(z)\,\mathrm{d}z\right).$$

When we evaluate the gradient at $r(z) = q_\phi(z)$, the right-most term vanishes, since $\int \frac{\nabla_\phi q_\phi(z)}{r(z)}r(z)\,\mathrm{d}z = \int \nabla_\phi q_\phi(z)\,\mathrm{d}z = 0$. Thus, the gradient of the log-variance loss becomes equal to the gradient of the KL divergence. $\square$

### A.2 Proof of Lemma 1

*Proof.* First, notice that $\text{Var}_{q_\phi}(\partial_{\phi_i}\log q_\phi) = \mathbb{E}_{q_\phi}[(\partial_{\phi_i}\log q_\phi)^2]$ since $\mathbb{E}_{q_\phi}[\partial_{\phi_i}\log q_\phi] = 0$. We then compute

$$a_i^* = \frac{\mathbb{E}_{q_\phi}\left[f_\phi(\partial_{\phi_i}\log q_\phi)^2\right]}{\mathbb{E}_{q_\phi}\left[(\partial_{\phi_i}\log q_\phi)^2\right]} \qquad (23)$$

$$= \frac{\mathbb{E}_{q_\phi}\left[f_\phi(\partial_{\phi_i}\log q_\phi)^2\right] - \mathbb{E}_{q_\phi}\left[f_\phi\right]\mathbb{E}_{q_\phi}\left[(\partial_{\phi_i}\log q_\phi)^2\right] + \mathbb{E}_{q_\phi}\left[f_\phi\right]\mathbb{E}_{q_\phi}\left[(\partial_{\phi_i}\log q_\phi)^2\right]}{\mathbb{E}_{q_\phi}\left[(\partial_{\phi_i}\log q_\phi)^2\right]} \qquad (24)$$

$$= \mathbb{E}_{q_\phi}[\bar{f}_\phi] + \delta_i^{\text{CV}}. \qquad (25)$$

In the last line we have used the fact that $\mathbb{E}_{q_\phi}[f_\phi] = \mathbb{E}_{q_\phi}[\bar{f}_\phi]$. $\square$

### A.3 Proof of Proposition 2

*Proof.* Note that

$$\left|\frac{\delta_i^{\text{CV}}}{\mathbb{E}_{q_\phi}[a^{\text{VarGrad}}]}\right| = \left|\frac{\text{Cov}_{q_\phi}\left(f_\phi, (\partial_{\phi_i}\log q_\phi)^2\right)}{\mathbb{E}_{q_\phi}[f_\phi]\text{Var}_{q_\phi}(\partial_{\phi_i}\log q_\phi)}\right| = \left|\frac{\mathbb{E}_{q_\phi}\left[(f_\phi - \mathbb{E}_{q_\phi}[f_\phi])(\partial_{\phi_i}\log q_\phi)^2\right]}{\mathbb{E}_{q_\phi}[f_\phi]\mathbb{E}_{q_\phi}[(\partial_{\phi_i}\log q_\phi)^2]}\right|, \qquad (26)$$

where we have used the fact that $\mathbb{E}_{q_\phi}[\partial_{\phi_i}\log q_\phi] = 0$. From

$$\mathbb{E}[f_\phi] = -\text{ELBO}(\phi) = \text{KL}(q_\phi(z) \,||\, p(z|x)) - \log p(x),$$

and using the Cauchy-Schwarz inequality, Eq. 26 can be bounded from above by

$$\frac{\left(\text{Var}_{q_\phi}\left(\log\frac{q_\phi(z)}{p(z|x)}\right)\right)^{1/2}}{|\text{KL}(q_\phi(z) \,||\, p(z|x)) - \log p(x)|}\left(\frac{\mathbb{E}_{q_\phi}[(\partial_{\phi_i}\log q_\phi)^4]}{(\mathbb{E}_{q_\phi}[(\partial_{\phi_i}\log q_\phi)^2])^2}\right)^{1/2}. \qquad (27)$$

The second factor equals $\sqrt{\text{Kurt}[\partial_{\phi_i}\log q_\phi]}$. To bound the first factor, notice that

$$\left(\text{Var}_{q_\phi}\left(\log\frac{q_\phi(z)}{p(z|x)}\right)\right)^{1/2} \leq \left(\mathbb{E}_{q_\phi}\left[\log^2\frac{q_\phi(z)}{p(z|x)}\right]\right)^{1/2} = \left(\mathbb{E}_{q_\phi}\left[\log^2\frac{p(z|x)}{q_\phi(z)}\right]\right)^{1/2} \qquad (28)$$

$$\leq \left(2\mathbb{E}_{q_\phi}\left[\exp\left(\left|\log\frac{p(z|x)}{q_\phi(z)}\right|\right) - 1 - \left|\log\frac{p(z|x)}{q_\phi(z)}\right|\right]\right)^{1/2}, \qquad (29)$$

where we have used the estimate

$$e^x - 1 - x = \sum_{n=0}^{\infty} \frac{x^n}{n!} - 1 - x = \sum_{n=2}^{\infty} \frac{x^n}{n!} \geq \frac{1}{2}x^2, \quad x \geq 0, \tag{30}$$

with $x = \left| \log \frac{p(z|x)}{q_\phi(z)} \right|$. We now use [Ghosal et al., 2000, Lemma 8.3] to bound Eq. 29 from above by

$$2\sqrt{C}h(q_\phi(z) \,||\, p(z|x)), \tag{31}$$

where

$$h(q_\phi(z) \,||\, p(z|x)) = \sqrt{\int \left( \sqrt{q_\phi(z)} - \sqrt{p(z|x)} \right)^2 \mathrm{d}z} \tag{32}$$

is the Hellinger distance. From [Reiss, 2012, Lemma A.3.5] we have the bound $h(q_\phi(z) \,||\, p(z|x)) \leq \sqrt{\mathrm{KL}(q_\phi(z) \,||\, p(z|x))}$. Combining these estimates we arrive at the claimed result. $\square$

## A.4  Proof of Proposition 3

*Proof.* We start by defining the short-cuts

$$A = f_\phi(z), \qquad B = (\partial_{\phi_i} \log q_\phi)(z). \tag{33}$$

Let us compute the difference of the variances of the estimators to leading order in $S$, namely

$$\mathrm{Var}(\widehat{g}_{\mathrm{Reinforce},i}) - \mathrm{Var}(\widehat{g}_{\mathrm{VarGrad},i}) = \frac{1}{S}\mathrm{Var}(AB) + \frac{S-2}{S(S-1)}\mathbb{E}\left[(A - \mathbb{E}[A])(B - \mathbb{E}[B])\right]^2 \tag{34a}$$

$$- \frac{\mathrm{Var}(A)\mathrm{Var}(B)}{S(S-1)} - \frac{1}{S}\mathbb{E}\left[(A - \mathbb{E}[A])^2(B - \mathbb{E}[B])^2\right] \tag{34b}$$

$$= \frac{1}{S}\left(\mathbb{E}\left[A^2B^2\right] - \mathbb{E}[AB]^2\right) + \frac{S-2}{S(S-1)}\mathbb{E}[AB]^2 \tag{34c}$$

$$- \frac{1}{S}\left(\mathbb{E}\left[A^2B^2\right] - 2\mathbb{E}[A]\mathbb{E}\left[AB^2\right] + \mathbb{E}[A]^2\mathbb{E}\left[B^2\right]\right) + \mathcal{O}\left(\frac{1}{S^2}\right) \tag{34d}$$

$$= -\frac{1}{S(S-1)}\mathbb{E}[AB]^2 - \frac{1}{S}\mathbb{E}[A]\left(\mathbb{E}[A]\mathbb{E}\left[B^2\right] - 2\mathbb{E}\left[AB^2\right]\right) + \mathcal{O}\left(\frac{1}{S^2}\right) \tag{34e}$$

$$= \frac{1}{S}\mathbb{E}[A]\left(2\mathbb{E}\left[AB^2\right] - \mathbb{E}[A]\mathbb{E}\left[B^2\right]\right) + \mathcal{O}\left(\frac{1}{S^2}\right) \tag{34f}$$

$$= \frac{1}{S}\mathbb{E}[A]\mathbb{E}\left[B^2\right]\left(2\delta_i^{\mathrm{CV}} + \mathbb{E}[A]\right) + \mathcal{O}\left(\frac{1}{S^2}\right) \tag{34g}$$

and we note that with $\mathbb{E}\left[B^2\right] > 0$ the leading term is positive if

$$\mathbb{E}[A]\delta_i^{\mathrm{CV}} + \frac{1}{2}\mathbb{E}[A]^2 > 0, \tag{34h}$$

which is equivalent to the statement in the proposition. $\square$

## A.5  Proof of Corollary 1

*Proof.* Note that with Proposition 2 we have

$$\left| \frac{\delta_i^{\mathrm{CV}}}{\mathbb{E}_{q_\phi}[a^{\mathrm{VarGrad}}]} \right| \to 0 \tag{35}$$

for $D \to \infty$, assuming that $\mathrm{KL}(q_\phi(z) \,||\, p(z\,|\,x))$ is strictly increasing in $D$. Therefore, for large enough $D$, the condition from Proposition 3 (see Eq. 19), is fulfilled and the statement follows immediately. $\square$

## A.6 Results on the Kurtosis of the Score for Exponential Families

Here we provide a more explicit expression for $\text{Kurt}[\partial_{\phi_i} \log q_\phi]$ in the case when $q_\phi(z)$ is given by an exponential family, i.e. $q_\phi(z) = h(z) \exp\left(\phi^\top T(z) - A(\phi)\right)$, where $T(z)$ is the vector of sufficient statistics and $A(\phi)$ denotes the log-partition function. As an application, we show that in the Gaussian case, $\text{Kurt}[\partial_{\phi_i} \log q_\phi]$ is uniformly bounded across the whole variational family.

**Lemma 2.** *Let $q_\phi(z) = h(z) \exp\left(\phi^\top T(z) - A(\phi)\right)$. Then*

$$\text{Kurt}[\partial_{\phi_i} \log q_\phi] = \frac{\mathbb{E}_{q_\phi}\left[(T_i(z) - m_i)^4\right]}{\mathbb{E}_{q_\phi}\left[(T_i(z) - m_i)^2\right]^2}, \tag{36}$$

*where $m_i = \mathbb{E}_{q_\phi}[T_i(z)]$ denotes the mean of the sufficient statistics. In particular, $\text{Kurt}[\partial_{\phi_i} \log q_\phi]$ does not depend on $h(z)$ or $A(\phi)$.*

*Proof.* The claim follows by direct calculation. Indeed,

$$\partial_{\phi_i} \log q_\phi(z) = T_i(z) - \frac{\partial A}{\partial \phi_i}(\phi). \tag{37}$$

It is left to show that $\frac{\partial A}{\partial \phi_i}(\phi) = \mu_i$. For this, notice that the normalisation condition

$$\int h(z) \exp\left(\phi^\top T(z) - A(\phi)\right) \mathrm{d}z = 1 \tag{38}$$

implies

$$\int h(z) \left(T_i(z) - \frac{\partial A(\phi)}{\partial \phi_i}\right) \exp\left(\phi^\top T(z) - A(\phi)\right) \mathrm{d}z = 0 \tag{39}$$

by taking the derivative w.r.t. $\phi_i$. The left-hand side equals $\mathbb{E}_{q_\phi}[T_i(z)] - \frac{\partial A}{\partial \phi_i}(\phi)$, and so the claim follows. $\square$

**Lemma 3.** *Let $q_\phi(z)$ be the family of one-dimensional Gaussian distributions. Then there exists a constant $K > 0$ such that*

$$\text{Kurt}[\partial_{\phi_i} \log q_\phi] < K \tag{40}$$

*for all $i$ and all $\phi \in \Phi$. In fact, it is possible to take $K = 15$.*

*Proof.* For the Gaussian family, the sufficient statistics are given by $T_1(z) = z$ and $T_2(z) = z^2$. We have that

$$\frac{\mathbb{E}_{\mathcal{N}(\mu,\sigma^2)}\left[(T_1(z) - m_1)^4\right]}{\mathbb{E}_{\mathcal{N}(\mu,\sigma^2)}\left[(T_1(z) - m_1)^2\right]^2} = \frac{\mathbb{E}_{\mathcal{N}(\mu,\sigma^2)}\left[(z - \mu)^4\right]}{\mathbb{E}_{\mathcal{N}(\mu,\sigma^2)}\left[(z - \mu)^2\right]^2} = 3, \tag{41}$$

by the well-known fact the standard kurtosis of any univariate Gaussian is 3. A lengthy but straightforward computation shows that

$$\frac{\mathbb{E}_{\mathcal{N}(\mu,\sigma^2)}\left[(T_2(z) - m_2)^4\right]}{\mathbb{E}_{\mathcal{N}(\mu,\sigma^2)}\left[(T_2(z) - m_2)^2\right]^2} = \frac{3(4\mu^4 + 20\mu^2\sigma^2 + 5\sigma^4)}{(2\mu^2 + \sigma^2)^2}, \tag{42}$$

which is maximised for $\mu = 0$, taking the value 15. $\square$

Lemma 3 shows that the kurtosis term in our bound Eq. 16 can be bounded for Gaussian families. This result is expected to extend to the multivariate cases as well. We note that we observe in our experiments that the bound is finite in a variety of cases.

## A.7 Dimension-dependence of the KL-divergence

The following lemma shows that the KL-divergence increases with the number of dimensions. This result follows from the chain-rule of KL divergence, see, e.g., Cover and Thomas [2012].

**Lemma 4.** *Let $u^{(D)}(z_1, \ldots, z_D)$ and $v^{(D)}(z_1, \ldots, z_D)$ be two arbitrary probability distributions on $\mathbb{R}^D$. For $J \in \{1 \ldots, D\}$ denote their marginals on the first $J$ coordinates by $u^{(J)}$ and $v^{(J)}$, i.e.*

$$u^{(J)}(z_1, \ldots, z_J) = \int \cdots \int u^{(D)}(z_1, \ldots, z_D) \, \mathrm{d}z_{J+1} \ldots \mathrm{d}z_D, \tag{43}$$

*and*

$$v^{(J)}(z_1, \ldots, z_J) = \int \cdots \int v^{(D)}(z_1, \ldots, z_D) \, \mathrm{d}z_{J+1} \ldots \mathrm{d}z_D. \tag{44}$$

*Then*

$$\mathrm{KL}(u^{(1)} \,||\, v^{(1)}) \leq \mathrm{KL}(u^{(2)} \,||\, v^{(2)}) \leq \ldots \leq \mathrm{KL}(u^{(D)} \,||\, v^{(D)}), \tag{45}$$

*i.e. the function $J \mapsto \mathrm{KL}(u^{(J)} \,||\, v^{(J)})$ is increasing.*

# B  Details of Experimental Setup and Additional Results

## B.1  Details of the Experiments

### B.1.1  Logistic Regression

This section describes the experimental setup of the Bayesian logistic regression example which was discussed in the main text in Section 6.

**Data.**  We use a synthetic dataset with $N = 100$, where input-output pairs are generated as follows: we sample a design matrix $X \in \mathbb{R}^{N \times D}$ for the inputs uniformly on $[-1, 1]$, random weights $\mathbf{w} \in \mathbb{R}^D$ from $\mathcal{N}(0, 25 \, \mathrm{Id}_{D \times D})$ and a random bias $b \in \mathbb{R}$ from $\mathcal{N}(0, 1)$. We set $\mathbf{p} = \sigma(X\mathbf{w} + b\,\mathbf{1})$, where $\mathbf{1}$ is an $N$-dimensional vector of ones and $\sigma(x) = \frac{1}{1+\exp(-x)}$ is the logistic sigmoid applied elementwise. Finally, we sample the outputs $Y \sim \mathrm{Bernoulli}(\mathbf{p})$.

**Approximate Posterior.**  For this model, we set the approximate posterior to a diagonal Gaussian with free mean and log standard deviation parameters.

**Training.**  For all the experiments listed in the main text, we use the VarGrad estimator for the gradients of the logistic regression models. We train the models using stochastic gradient descent [Robbins and Monro, 1951] with a learning rate of 0.001.

**Estimation of intractable quantities.**  To estimate the intractable quantities in Figure 1, we use Monte Carlo sampling with 2000 samples for $\delta^{\mathrm{CV}}$ and $\mathbb{E}_{q_\phi}[a^{\mathrm{VarGrad}}]$. We estimate the KL divergence with the identity $\mathrm{KL}(q_\phi(z)\|p(z|x)) = \log p(x) - \mathrm{ELBO}(\phi)$, where $\log p(x)$ is estimated using importance sampling with 10000 samples and $\mathrm{ELBO}(\phi)$ using standard Monte Carlo sampling with 2000 samples.

For the variance estimates in Figure 3, we use 1000 Monte Carlo samples. As explained in the main text, to estimate the control variate coefficients, we use either 2 samples for the *sampled estimator* or 1000 samples for the *oracle estimator*.

## B.2  Discrete VAEs

This section describes the experimental setting for the Discrete VAE, where we closely follow the setup in Maddison et al. [2017], which was also replicated in Tucker et al. [2017] and Grathwohl et al. [2018]. As we are comparing the usefulness of different estimators in the optimisation and time their run-times, we opted to re-implement the various methods using JAX [Bradbury et al., 2018, Hennigan et al., 2020]. Extra care was taken to be as faithful as possible to the implementation description in the respective papers as well as in optimising the run-time of the implementations.

**Data.**  We use a fixed binarisation of Omniglot [Lake et al., 2015], where we binarise at the standard cut-off of 0.5. We use the standard train/test splits for this dataset.

**Model Architectures.**  For the DVAE experiments we use the *two layers linear* architecture, which has 2 stochastic binary layers with 200 units each, which was used in Maddison et al. [2017]. For this model, the decoders mirror the corresponding encoders. We use a Bernoulli(0.5) prior on the latent space and fix its parameters throughout the optimisation.

**Approximate Posterior.**  We use an amortised mean-field Bernoulli approximation for the posterior.

**Training.**  For training the models, we use the Adam optimiser [Kingma and Ba, 2015] with learning rates 0.001, 0.0005 and 0.0001.

**Estimation of intractable quantities** .  We use Monte Carlo sampling with 2000 samples for $\delta^{\mathrm{CV}}$ and $\mathbb{E}_{q_\phi}[a^{\mathrm{VarGrad}}]$. Due to the high memory requirements of these computations and sparsity of the weight gradients, we only compute them for the biases. To estimate the gradient variances we use 1000 Monte Carlo samples.

### B.3 Additional Results for DVAEs

#### B.3.1 Variance Reduction

In Figure B.5 we present additional results on the practical variance reduction that VarGrad induces in the two layer linear DVAE. Here, we compare with various other estimators from the literature. VarGrad achieves considerable variance reduction over the adaptive (RELAX) and non-adaptive (Controlled Reinforce) model-agnostic estimators. Structured adaptive estimators such as Dynamic REBAR and RELAX + REBAR start with a higher variance at the beginning of optimisation, which reduces towards the end. ARM, which uses antithetic sampling, achieves the most reduction; however, it is only applicable to models with Bernoulli latent variables. Notably, the extra variance reduction seen in some of the methods does not translate to better optimisation performance on this example as seen in Appendix B.3.2.

*Figure B.5.* Estimates of the gradient variance of the DVAE at 4 points during the optimisation for different gradient estimators. The plot compares VarGrad to Reinforce with score function control variates [Ranganath et al., 2014], dynamic REBAR [Tucker et al., 2017], RELAX, RELAX + REBAR [Grathwohl et al., 2018] and ARM [Yin and Zhou, 2019]. The number of samples used to compute each gradient estimator is given in the figure legend.

#### B.3.2 Performance in Optimisation

In this section we present additional results on training the DVAE with VarGrad. Figure B.6 replicates Figure 4 with a longer run-time. Figure B.7 and Figure B.8 show the optimisation traces for different Adam learning rates.

*Figure B.6.* Optimisation trace versus epoch (left) and wall-clock time (right) for a two-layer linear DVAE on a fixed binarisation of Omniglot, trained with Adam with a learning rate of 0.001. The plot compares VarGrad to Reinforce with score function control variates [Ranganath et al., 2014], dynamic REBAR [Tucker et al., 2017], RELAX, RELAX + REBAR [Grathwohl et al., 2018] and ARM [Yin and Zhou, 2019]. The number of samples used to compute each gradient estimator is given in the figure legend. The results here are identical to the ones in Figure 4 but with a longer run-time.

*Figure B.7.* Optimisation trace versus epoch (left) and wall-clock time (right) for a two-layer linear DVAE on a fixed binarisation of Omniglot, trained with Adam with a learning rate of 0.0005. The plot compares VarGrad to Reinforce with score function control variates [Ranganath et al., 2014], dynamic REBAR [Tucker et al., 2017], RELAX, RELAX + REBAR [Grathwohl et al., 2018] and ARM [Yin and Zhou, 2019]. The number of samples used to compute each gradient estimator is given in the figure legend.

*Figure B.8.* Optimisation trace versus epoch (left) and wall-clock time (right) for a two-layer linear DVAE on a fixed binarisation of Omniglot, trained with Adam with a learning rate of 0.0001. The plot compares VarGrad to Reinforce with score function control variates [Ranganath et al., 2014], dynamic REBAR [Tucker et al., 2017], RELAX, RELAX + REBAR [Grathwohl et al., 2018] and ARM [Yin and Zhou, 2019]. The number of samples used to compute each gradient estimator is given in the figure legend.

## C Gaussians

In the case when $q(z)$ and $p(z|x)$ are (diagonal) Gaussians we can gain some intuition on the performance of VarGrad by computing the relevant quantities analytically. The principal insights obtained from the examples presented in this section can be summarised as follows: Firstly, in certain scenarios the Reinforce estimator does indeed exhibit a lower variance in comparison with VarGrad (although the advantage is very modest and only materialises for a restricted set of parameters). This finding illustrates that the conditions in Eq. 19 and Eq. 20 (the latter referring to $S \geq S_0$) cannot be dropped without replacement from the formulation of Proposition 3. Secondly, in line with the results from Section 6, the relative error $\delta_i^{\mathrm{CV}}/\mathbb{E}[a^{\mathrm{VarGrad}}]$ decreases with increased dimensionality. Moreover, the variance associated to computing the optimal control variate coefficients $a^*$ is significant and increases considerably with the number of latent variables.

### C.1 Comparing the Variances of Reinforce and VarGrad

In order to understand when the variance of VarGrad is smaller than the variance of the Reinforce estimator we first consider the one-dimensional Gaussian case $q(z) = \mathcal{N}(z; \mu, \sigma^2)$ and $p(z|x) = \mathcal{N}(z; \widetilde{\mu}, \widetilde{\sigma}^2)$ and analyse the derivative w.r.t. $\mu$. A lengthy calculation shows that

$$\Delta_{\mathrm{Var}}(\mu, \widetilde{\mu}, \sigma^2, \widetilde{\sigma}^2, S) := \mathrm{Var}(\widehat{g}_{\mathrm{Reinforce},\mu}) - \mathrm{Var}(\widehat{g}_{\mathrm{VarGrad},\mu}) \tag{46a}$$

$$= \frac{1}{4S\sigma^4\widetilde{\sigma}^2} \left( (\mu - \widetilde{\mu})^4 + 2(\mu - \widetilde{\mu})^2 \left( \frac{3S-7}{S-1}\sigma^2 - 3\widetilde{\sigma}^2 \right) + \frac{5S-7}{S-1} \left( \sigma^2 - \widetilde{\sigma}^2 \right)^2 \right) \tag{46b}$$

$$\approx \frac{1}{4S\sigma^4\widetilde{\sigma}^2} \left( \Delta_\mu^4 + 6\Delta_\mu^2 \Delta_{\sigma^2} + 5\Delta_{\sigma^2}^2 \right), \tag{46c}$$

where the last line holds for large $S$ with $\Delta_\mu := \mu - \widetilde{\mu}$ and $\Delta_{\sigma^2} := \sigma^2 - \widetilde{\sigma}^2$.

For an illustration, let us vary the above parameters. First, let us fix $\sigma^2 = \widetilde{\sigma}^2 = 1$. We note from Eq. 46c that in this case we expect VarGrad to have lower variance regardless of $\Delta_\mu$ as long as $S$ is large enough. In Figure C.9 we see that this is in fact the case, however a different result can be observed for small $S$, which is again in accordance with Eq. 46b.

Next, we consider arbitrary $\sigma^2$ and $\widetilde{\sigma}^2$, but fixed $\mu = 1, \widetilde{\mu} = 2$. In Figure C.10 we observe that the variance of VarGrad is smaller for most values of $\sigma^2$ and $\widetilde{\sigma}^2$. However, even for large $S$ there remains a region where the Reinforce estimator is superior. In fact, one can compute the condition for this to happen to be $\Delta_{\sigma^2} \in \left[ -\Delta_\mu^2, -\frac{1}{5}\Delta_\mu^2 \right]$, which can be compared with the condition in Eq. 19 in Proposition 3.

In Figure C.11 we display the variance differences $\Delta_{\mathrm{Var}}$ as functions of $\Delta_\mu$ and $\Delta_{\sigma^2}$, approximated according to Eq. 46c, for the same fixed values as before and see that they are bounded from below, but not from above.

For a $D$-dimensional Gaussian it is hard to compute the condition from Eq. 19 in full generality, but we can derive the following stronger criterion that can guarantee better performance of VarGrad when assuming that $\mathrm{ELBO}(\phi) \leq 0$ (which for instance holds in the discrete-data setting).

**Lemma 5.** *Assume* $\mathrm{ELBO}(\phi) \leq 0$ *and*

$$\mathrm{Cov}\left( f_\phi, (\partial_{\phi_i} \log q_\phi)^2 \right) > 0. \tag{47}$$

*Then there exists* $S_0 \in \mathbb{N}$ *such that*

$$\mathrm{Var}\left( \widehat{g}_{VarGrad,i}(\phi) \right) \leq \mathrm{Var}\left( \widehat{g}_{Reinforce,i}(\phi) \right), \qquad \text{for all} \quad S \geq S_0. \tag{48}$$

*Proof.* With $\mathrm{ELBO}(\phi) \leq 0$ we have

$$\mathrm{Cov}\left( f_\phi, (\partial_{\phi_i} \log q_\phi)^2 \right) \leq \mathbb{E}_{q_\phi}\left[ f_\phi (\partial_{\phi_i} \log q_\phi)^2 \right] - \frac{1}{2} \mathbb{E}_{q_\phi}[f_\phi] \mathbb{E}_{q_\phi}\left[ (\partial_{\phi_i} \log q_\phi)^2 \right] \tag{49}$$

$$= \mathbb{E}_{q_\phi}\left[ (\partial_{\phi_i} \log q_\phi)^2 \right] \left( \delta_i^{\mathrm{CV}} - \frac{1}{2}\mathrm{ELBO}(\phi) \right). \tag{50}$$

If now

$$\mathrm{Cov}\left( f_\phi, (\partial_{\phi_i} \log q_\phi)^2 \right) > 0, \tag{51}$$

*Figure C.9.* We compare the variance of the reinforce estimator with the variance of VarGrad. VarGrad is often better even for small $S$ – for large $S$ this can be guaranteed with Proposition 3.

then also

$$\delta_i^{\text{CV}} - \frac{1}{2}\text{ELBO}(\phi) > 0, \tag{52}$$

and the statement follows by Proposition 3. $\square$

The condition from Eq. 47 gives another guarantee for VarGrad having smaller variance than the Reinforce estimator. However, we note that the converse statement is not necessarily true, i.e. if the condition does not hold, VarGrad can still be better. The advantage of Eq. 47, however, is that it can be verified more easily in certain settings, as for instance done for $D$-dimensional diagonal Gaussians in the following lemma.

**Lemma 6** (Covariance term for diagonal Gaussians)**.** *Let $q(z)$ and $p(z|x)$ be diagonal $D$-dimensional Gaussians with means $\mu$ and $\widetilde{\mu}$ and covariance matrices $\Sigma = \text{diag}(\sigma_1^2, \ldots, \sigma_D^2)$ and $\widetilde{\Sigma} = \text{diag}(\widetilde{\sigma}_1^2, \ldots, \widetilde{\sigma}_D^2)$. Then*

$$\text{Cov}_{q_\phi}\left(f_\phi, (\partial_{\phi_k} \log q_\phi)^2\right) = \frac{1}{\widetilde{\sigma}_k^2} - \frac{1}{\sigma_k^2} \tag{53}$$

*for $k \in \{1, \ldots, D\}$ and*

$$\text{Cov}_{q_\phi}\left(f_\phi, (\partial_{\phi_k} \log q_\phi)^2\right) = \frac{1}{\sigma_k^2}\left(\frac{1}{\widetilde{\sigma}_k^2} - \frac{1}{\sigma_k^2}\right) \tag{54}$$

*for $k \in \{D+1, \ldots, 2D\}$ with $\phi = (\mu_1, \ldots, \mu_D, \sigma_1^2, \ldots, \sigma_D^2)^\top$ and*

$$\text{Cov}_{q_\phi}\left(f_\phi, (\partial_{\phi_k} \log q_\phi)^2\right) = \frac{1}{\widetilde{\sigma}_k^2} - \frac{1}{\sigma_k^2} \tag{55}$$

*for $k \in \{D+1, \ldots, 2D\}$ with $\phi = (\mu_1, \ldots, \mu_D, \log \sigma_1^2, \ldots, \log \sigma_D^2)^\top$.*

*Proof.* We compute

$$f_\phi = -\frac{1}{2}\sum_{i=1}^{D} \log\left(\frac{\sigma_i^2}{\widetilde{\sigma}_i^2}\right) - \frac{1}{2}\sum_{i=1}^{D}\frac{(z_i - \mu_i)^2}{\sigma_i^2} + \frac{1}{2}\sum_{i=1}^{D}\frac{(z_i - \widetilde{\mu}_i)^2}{\widetilde{\sigma}_i^2} \tag{56}$$

*Figure C.10.* Variance comparison with varying $\sigma^2$ and $\widetilde{\sigma}^2$. VarGrad only wins outside a certain region, however, if so, then potentially by orders of magnitude.

*Figure C.11.* Variance differences of the reinforce estimator and VarGrad with varying $\Delta_\mu$ and $\Delta_{\sigma^2}$ for different sample sizes $S$.

and

$$\partial_{\mu_k} \log q_\phi = \frac{z_k - \mu_k}{\sigma_k^2}. \tag{57}$$

We again use the short-cuts

$$A = f_\phi(z), \qquad B = \left(\partial_{\mu_k} \log q_\phi\right)(z), \tag{58}$$

and obtain

$$\text{Cov}_{q_\phi}(A, B^2) = \mathbb{E}_{q_\phi}\left[AB^2\right] - \mathbb{E}_{q_\phi}[A]\,\mathbb{E}_{q_\phi}\left[B^2\right] \tag{59}$$

$$= \mathbb{E}_{q_\phi}\left[\left(-\frac{1}{2}\sum_{i=1}^{D}\log\left(\frac{\sigma_i^2}{\widetilde{\sigma}_i^2}\right) - \frac{1}{2}\sum_{i=1}^{D}\frac{(z_i-\mu_i)^2}{\sigma_i^2} + \frac{1}{2}\sum_{i=1}^{D}\frac{(z_i-\widetilde{\mu}_i)^2}{\widetilde{\sigma}_i^2}\right)\left(\frac{z_k-\mu_k}{\sigma_k^2}\right)^2\right] \tag{60}$$

$$- \mathbb{E}_{q_\phi}\left[\left(-\frac{1}{2}\sum_{i=1}^{D}\log\left(\frac{\sigma_i^2}{\widetilde{\sigma}_i^2}\right) - \frac{1}{2}\sum_{i=1}^{D}\frac{(z_i-\mu_i)^2}{\sigma_i^2} + \frac{1}{2}\sum_{i=1}^{D}\frac{(z_i-\widetilde{\mu}_i)^2}{\widetilde{\sigma}_i^2}\right)\right]\mathbb{E}_{q_\phi}\left[\left(\frac{z_k-\mu_k}{\sigma_k^2}\right)^2\right]. \tag{61}$$

$$= -\frac{1}{2}\left(\frac{3}{\sigma_k^2} + \frac{D-1}{\sigma_k^2}\right) + \frac{1}{2}\left(\frac{1}{\sigma_k^2}\sum_{\substack{i=1\\i\neq k}}^{D}\frac{\sigma_i^2+(\mu_i-\widetilde{\mu}_i)^2}{\widetilde{\sigma}_i^2} + \frac{1}{\sigma_k^2\widetilde{\sigma}_k^2}\left(3\sigma_k^2+(\mu_k-\widetilde{\mu}_k)^2\right)\right) \tag{62}$$

$$- \left(-\frac{D}{2} + \frac{1}{2}\sum_{i=1}^{D}\frac{\sigma_i^2+(\mu_i-\widetilde{\mu}_i)^2}{\widetilde{\sigma}_i^2}\right)\frac{1}{\sigma_k^2} \tag{63}$$

$$= -\frac{1}{\sigma_k^2} + \frac{1}{2\sigma_k^2\widetilde{\sigma}_k^2}\left(3\sigma_k^2+(\mu-\widetilde{\mu})^2\right) - \frac{1}{2\sigma_k^2\widetilde{\sigma}_k^2}\left(\sigma_k^2+(\mu-\widetilde{\mu})^2\right) \tag{64}$$

$$= \frac{1}{\widetilde{\sigma}_k^2} - \frac{1}{\sigma_k^2}. \tag{65}$$

For the terms with the partial derivative w.r.t. $\sigma_k^2$ we first note that

$$\partial_{\sigma_k^2}\log q_\phi = -\frac{1}{2\sigma_k^2} + \frac{(z_k-\mu_k)^2}{2\sigma_k^4} = \frac{1}{\sigma_k^2}\partial_{\log\sigma_k^2}\log q_\phi. \tag{66}$$

We compute

$$\mathbb{E}_{q_\phi}\left[AB^2\right] = \mathbb{E}_{q_\phi}\left[\frac{(z_k-\mu_k)^2}{4\sigma_k^6}\sum_{i=1}^{D}\frac{(z_i-\mu_i)^2}{\sigma_i^2} - \frac{(z_k-\mu_k)^4}{8\sigma_k^8}\sum_{i=1}^{D}\frac{(z_i-\mu_i)^2}{\sigma_i^2}\right. \tag{67}$$

$$\left. - \frac{(z_k-\mu_k)^2}{4\sigma_k^6}\sum_{i=1}^{D}\frac{(z_i-\widetilde{\mu}_i)^2}{\widetilde{\sigma}_i^2} + \frac{(z_k-\mu_k)^4}{8\sigma_k^8}\sum_{i=1}^{D}\frac{(z_i-\widetilde{\mu}_i)^2}{\widetilde{\sigma}_i^2}\right] \tag{68}$$

$$= -\frac{1}{8\sigma_k^4}(D+8) + \frac{1}{8\sigma_k^4\widetilde{\sigma}_k^2}\left(9\sigma_k^2+(\mu_k-\widetilde{\mu}_k)^2\right) + \frac{1}{8\sigma_k^4}\sum_{\substack{i=1\\i\neq k}}^{d}\frac{\sigma_i^2+(\mu_i-\widetilde{\mu}_i)^2}{\widetilde{\sigma}_i^2}, \tag{69}$$

and similarly

$$\mathbb{E}_{q_\phi}[A]\,\mathbb{E}_{q_\phi}\left[B^2\right] = -\frac{D}{8\sigma_k^4} + \frac{1}{8\sigma_k^4\widetilde{\sigma}_k^2}\left(\sigma_k^2+(\mu_k-\widetilde{\mu}_k)^2\right) + \frac{1}{8\sigma_k^4}\sum_{\substack{i=1\\i\neq k}}^{D}\frac{\sigma_i^2+(\mu_i-\widetilde{\mu}_i)^2}{\widetilde{\sigma}_i^2}. \tag{70}$$

We therefore get the result by again computing $\text{Cov}_{q_\phi}(A, B^2) = \mathbb{E}_{q_\phi}\left[AB^2\right] - \mathbb{E}_{q_\phi}[A]\,\mathbb{E}_{q_\phi}\left[B^2\right]$. The partial derivative w.r.t. $\log\sigma_k^2$ can be recovered from Eq. 66. □

## C.2 Optimal Control Variates in the Gaussian Case

In the diagonal Gaussian case we can also easily analytically compute the optimal control variate coefficients from Eq. 11, along the lines of the proof of Lemma 6. Our setting is again $q(z) = \mathcal{N}(z;\mu,\Sigma), p(z|x) = \mathcal{N}(z;\widetilde{\mu},\widetilde{\Sigma})$ with $\Sigma = \text{diag}\left(\sigma_1^2,\ldots,\sigma_D^2\right), \widetilde{\Sigma} = \text{diag}\left(\widetilde{\sigma}_1^2,\ldots,\widetilde{\sigma}_D^2\right)$. In Figure C.12 we plot the variances of four different gradient estimators with varying sample size $S$, namely $\widehat{g}_{\text{Reinforce}}, \widehat{g}_{\text{VarGrad}}$, as well as the Reinforce estimator augmented with the optimal control variate, once computed analytically and once sampled using $S$ samples. The variance depends on the mean and the covariance matrix; here we choose $\mu_i = 3, \sigma_i^2 = 3, \widetilde{\mu}_i = 1, \widetilde{\sigma}_i^2 = 1$ for all $i \in \{1,\ldots,D\}$. We observe that the VarGrad estimator is close to the analyitcal optimal control variate, and that the

*Figure C.12.* Comparion of the variances of the different gradient estimators $\widehat{g}_{\text{Reinforce}}$, $\widehat{g}_{\text{VarGrad}}$, as well as the reinforce estimator with the optimal control variate coefficient, once computed analytically and once sampled with $S$ samples, for dimensions $D = 3$ and $D = 30$.

*Figure C.13.* Mean, variance and relative errors associated to the two contributions to the optimal control variate coefficient, $\delta_i^{\text{CV}}$ and $a^{\text{VarGrad}} = \bar{f}_\phi$.

sampled optimal control variate performs significantly worse in a small sample size regime. These observations get more pronounced in higher dimensions and indicate that the variance of the sampled optimal control variate can itself be high, showing that using it might not always be beneficial in practice.

Let us additionally investigate the optimal control variate correction term $\delta_i^{\text{CV}}$ as defined in Eq. 13 for $D$-dimensional Gaussians $q(z)$ and $p(z|x)$ as considered above. In Figure C.13 we display the variances, means and relative errors of $\delta_i^{\text{CV}}$ and $a^{\text{VarGrad}} = \bar{f}_\phi$ and realise that indeed the ratio of those two converges to zero when $D$ gets larger. Furthermore we notice that the relative error of $\delta^{\text{CV}}$ increases with the dimension, explaining the difficulties when estimating the optimal control variate coefficients from samples. Finally, we plot a histogramm of $\delta_i^{\text{CV}}$ (varying across $i$) in Figure C.14, showing that $\delta_i^{\text{CV}}$ is small in comparison to $\mathbb{E}\left[a^{\text{VarGrad}}\right]$ and distributed around zero.

# D   Connections to Other Divergences

The Reinforce gradient estimator from Eq. 2 can as well be derived from the *moment loss*

$$\mathcal{L}_r^{\text{moment}}(q_\phi(z)\|p(z|x)) = \frac{1}{2}\,\mathbb{E}_{r(z)}\left[\log^2\left(\frac{q_\phi(z)}{p(z|x)}\right)\right], \tag{71}$$

namely

$$\nabla_\phi \mathcal{L}_r^{\text{moment}}(q_\phi(z)\|p(z|x))\Big|_{r=q_\phi} = \mathbb{E}_{q_\phi}\left[\log\left(\frac{q_\phi(z)}{p(z|x)}\right)\nabla_\phi \log q_\phi(z)\right]. \tag{72}$$

Histogram of $\delta_i^{\mathrm{CV}}$, diagonal Gaussian, $D = 350$

*Figure C.14.* The histogram of $\delta_i^{\mathrm{CV}}$ shows that it is usually rather small in comparison to $\mathbb{E}\left[a^{\mathrm{VarGrad}}\right]$, which is roughly 700 here, and that it fluctuates around zero.

In the log-variance loss, one can omit the logarithm to obtain the *variance loss*

$$\mathcal{L}_r^{\mathrm{Var}}(p(z|x)\|q_\phi(z)) = \frac{1}{2}\,\mathrm{Var}_{r(z)}\left(\frac{p(z|x)}{q_\phi(z)}\right) = \frac{1}{2}\,\mathbb{E}_{r(z)}\left[\left(\frac{p(z|x)}{q_\phi(z)}\right)^2 - 1\right], \tag{73}$$

which with $r = q_\phi$ coincides with the $\chi^2$-divergence. The potential of using the latter in the context of variational inference was suggested in Dieng et al. [2017]. We note that again one is in principle free in choosing $r(z)$, but that unlike the log-variance loss, this loss in not symmetric with respect to $q_\phi(z)$ and $p(z|x)$. An analysis in Nüsken and Richter [2020] (for distributions on path space) however suggests that the variance loss (unlike the log-variance loss) scales unfavourably in high-dimensional settings in terms of the variance associated to standard Monte Carlo estimators.