[Reviews · NeurIPS 2020]

Review 1

Summary and Contributions: This paper analyzes a gradient estimator for variational inference, herein dubbed VarGrad. The estimator -- already existing in the literature -- is derived from a new perspective which elucidates a simple implementation. This estimator is then analyzed both theoretically and empirically, demonstrating that it improves on the baseline REINFORCE estimator and remains competitive with other state-of-the-art gradient estimators, without requiring additional hyperparameter tuning.

Strengths: The theoretical analysis of the objective is novel and generally insightful. The simplicity of the new parametrization of the old gradient estimator is appealing. The paper is easy to understand but not trivial. The empirical evaluation specifically supports the theory and shows competitiveness with other gradient estimators. This paper is certainly relevant to the NeurIPS community; gradient estimators, particularly for discrete models, are a popular topic, and this paper demonstrates the theoretical and empirical benefits of a particular estimator while providing a simple implementation.

Weaknesses: The main weakness of this paper to me is that the gradient estimator studied in this paper was already known, which caps the novelty of the contribution. However, it is still important to develop a deeper understanding of pre-existing methods, and as such I do not think this lack of novelty is a huge concern. Another concern I have is that I am not totally clear on the class of approximate posterior distributions for which the theoretical results may be applicable. Lastly, since this paper only analyzes a previous method, we might want to have at least one more experimental setting comparing this estimator to others, or we might want to probe the theoretical claims on a more challenging example than one with synthetic data.

Correctness: The mathematical derivations all appear to be correct, and the methodology looks to be correct as well. The one concern I have about the correctness of claims is in Section 4. I am fine with the order of the error presented in the high- and low-KL regimes, but I have concerns that this order notation is hiding potentially large constants related to the Kurtosis for non-Gaussian distributions. Furthermore, I am not really sold on the significance of Corollary 1 - we have no idea how large $D_0$ needs to be to satisfy the limiting argument, and $D$ is *not* something that we can change in the method (as opposed to $S$). I would appreciate some clarity on these issues.

Clarity: I did find the paper to be quite well written.

Relation to Prior Work: The prior work section appears to be well-done, although I will note that I am not an expert on gradient estimators not based on reparametrization.

Reproducibility: Yes

Additional Feedback: *** POST AUTHOR FEEDBACK *** Having read the author response, the other reviews, and the discussion with other reviewers, I am inclined to keep my score at "Marginally above the acceptance threshold". *** PRE AUTHOR FEEDBACK *** I would first of all like to thank the authors for their paper. It appears to be a quite sound analysis of a previously-existing gradient estimator for variational inference, and it adds a novel implementation that is appealing simple for automatic differentiation engines. Since the paper is only analyzing a previous method, however, the theoretical and empirical evaluation should be held to a higher standard. I will focus a bit more in detail on both of these points below: - The theoretical evaluation was mostly good, as noted above. However, I am not sure what class of distributions satisfy the Kurtosis assumption of Proposition 2, and thus I am not totally convinced of the applicability of the result. This also plays into Proposition 3. - The empirical evaluation -- which was well done for the synthetic data example -- could have done a bit more to assuage these concerns, probing the correctness of the theoretical predictions in the MNIST example in the same way as in the previous example. Altogether, however, my assessment of the paper is a that it's a solid contribution which can be made better without too much effort, and I would be interested to hear the authors' response. Here are some smaller comments: - In the background section, some time is spent on how VI methods start by wanting to minimizie the KL divergence, but avoid computing the evidence by optimizing the ELBO instead. But then, we go back to taking gradients of the KL divergence directly. Why even bother to spend several lines talking about the ELBO here? I think the writing here could be streamlined. - It is mentioned that the log-variance loss is a divergence when the support of the reference contains the supports of q and p. I can see why this would be true, but is it worth providing a proof or reference? Is it possible that we might not be able to guarantee the equivalence of q and p on measure-zero sets of r? - L142: Inconsistent notation between z_s and z^(s) - L158: Inconsistent notation between \delta^{CV} and \delta_{CV} - Fig. 1: Can we add error bounds to this plot here? - L265: I am a bit confused by the language here. Is the oracle estimator using 1000 _extra_ samples, or just 1000 samples total? This is a minor point but the writing makes it a bit unclear.


Review 2

Summary and Contributions: The paper introduces a new stochastic gradient estimator for ELBO called VarGrad. Authors connect a special case of VarGrad to known estimators(Reinforce with Replacement). They analyse theoretically optimality of this special case of VarGrad in control variate sense. They show some empirical evidence for their theoretical predictions, as well as compare VarGrad performance to other estimators on toy dataset as well as Omniglot image dataset.

Strengths: Analysis of optimality of VarGrad is interesting and useful: it explains improved performance compared to Reinforce and gives a sense of how close VarGrad is to the optimal control variate.

Weaknesses: Authors didn't explore extensions of VarGrad beyond Reinforce with Replacement from (Kool et.al. (2019)), so effectively no new estimator was introduced. DVAE experiments with linear models are not sufficient to demonstrate utility of proposed estimator: non-linear models behave differently and it would be useful to add comparison for that case.

Correctness: The claims look correct and well justified, however adding experiments with non-linear DVAE seems necessary to show the advantage of VarGrad over other estimators.

Clarity: The paper is well-written

Relation to Prior Work: yes

Reproducibility: Yes

Additional Feedback: *******post rebuttal******* I'd like to thank the authors for their response and have increased my score to 5. I appreciate author's response about adding non-linear model results, however they did not add test ELBO performance which is of ultimate importance. Current ELBO performance plots are not very legible and aren't trained to convergence. I also find analysis of variance of limited use: in the general case it's not possible to quantify \delta^{CV} but in concrete cases other existing estimators are likely to perform better.


Review 3

Summary and Contributions: The paper considers a low variance gradient estimator for the ELBO objective used in variational inference. It consists in combining the standard Reinforce gradient estimator with Rao-Blackwellisation. The authors show that this estimator can be equivalently obtained as a multi-sample gradient estimator of a different objective function, namely the variance of the log ratio of the variational approximation q(z|x) and the true posterior p(z|x). Starting from this perspective, they analyse the properties of this estimator and its variance in comparison to the multi-sample Reinforce estimator.

Strengths: The theoretical part of paper is concise and sound. (However, I have not read all proofs in the supplementary materials). The proposed interpretation in terms of the variance objective opens a fresh perspective of the studied generic and low variance gradient estimator. Its applicability is not restricted to specific assumptions on the model posterior as e.g. differentiability w.r.t. the latent variables.

Weaknesses: The genericity of the studied gradient estimator is a weakness at the same time. It does not account for the specifics of the considered models and their structure. It will inevitably fall short of more specialised estimators designed for specific model classes as for instance deep networks. The newly proposed log-variance loss computes the divergence of two distributions against a third, arbitrary distribution. The authors have shown that fixing this distribution to the variational approximation q(z|x) after(!) gradient computation, reproduces the gradient of the ELBO objective. This is somewhat unsatisfying. I would have expected a deeper analysis here. For instance, can the maximisation w.r.t. to this probing distribution be seen as the dual of some distance measure that involves a minimisation (similar to Wasserstein distance)? The experimental validation presented in the paper is restricted to simple models. Moreover, the approach is compared with generic methods only. It remains unclear how it scales for deep networks.

Correctness: The theoretical claims seem to be correct. The experimental validation is in my view not complete.

Clarity: The paper is well structured and well written.

Relation to Prior Work: The paper discusses previous work and positions the proposed contributions w.r.t. to it.

Reproducibility: Yes

Additional Feedback: Post rebuttal: My main concerns about the analysis of the VarGrad estimator are: 1. Scalability of the approach to deep networks. 2. Deeper analysis w.r.t. the choice of the probing distribution r in the VarGrad estimator. The authors addressed them in their rebuttal only superficially. I therefore keep my "marginally above the acceptance threshold" recommendation.


Review 4

Summary and Contributions: *** POST AUTHOR FEEDBACK *** I keep my "marginally above the acceptance threshold" recommendation. If the authors can show that the proposed method works well for mixture approximations, I think the approach will be more applicable since existing methods are mainly tested on VAE-type models.  ------- In this work, the authors build a connection between the covariance estimator proposed by Salimans & Knowles 2014 and the log-variance loss proposed by Nusken & Richter 2020. The authors show that the proposed gradient estimator can be easily implemented via Auto-Diff and provide some analysis of the estimator.

Strengths: It has an impact on variational inference for discrete variables although the estimator is not new. It is equally important to bring back lesser-known gradient estimators if they perform well. The paper is well-written. The experimental results look good.  

Weaknesses: Experiments may be not enough. It will be great if some comparison study between re-parametrization gradients and the proposed method is included. (see the detailed comment)

Correctness: I think the method is sound. I do not check the correctness of the proofs since they are in the appendix

Clarity: The paper is well-written.

Relation to Prior Work: The log-variance loss is first proposed by Nusken & Richter 2020. It will be great if the authors mention the origin of the loss in Definition 1.

Reproducibility: Yes

Additional Feedback: For example, when a variational approximation is a finite mixture of Gaussians, it is challenging to learn the mixture weights.  Jankowiak & Karaletsos 2019 (Pathwise derivatives for multivariate distributions) propose a re-parametrization gradient for the mixing weights when diagonal Gaussian components are used.  To my best knowledge, there is no re-parametrization gradient method for finite mixture of Gaussians with full covariance structure. If the proposed method can work well in this case, it has an impact in the variational inference community beyond VAE.

[Author Response · NeurIPS 2020]

We thank all the reviewers for their stimulating comments and engaging questions. We are happy that they appreciated
our theoretical analyses and argued that those, as well as VarGrad's practical usefulness, are of benefit to the NeurIPS
community. We now address some questions and comments in the sequel.

*R1, R2: The gradient estimator was already known.* We agree and did not claim otherwise. We also appreciate that most
reviewers did not find this a concern. We emphasise that the novelty in our work is: (i) the derivation of this estimator
in terms of the log-variance loss, and (ii) the theoretical analysis of its variance. (i) This new perspective is simpler and
allows for a natural interpretation in terms of divergences (which is beneficial in other ML areas, e.g., reinforcement
learning). We also note that the connection with the log-variance loss is of practical interest, as it enables a simple
implementation algorithm based on automatic differentiation. Moreover, we believe this connection opens the door for
further research. (ii) Our work is the first to provide a theoretical analysis of the variance of this estimator. This analysis
shows that VarGrad's control variate coefficients are close to the optimal ones for the score function control variate.

*R1: What class of distributions satisfies the kurtosis condition of Proposition 2?* In general, the kurtosis will not
negatively affect the bound as long as the tails of the variational distribution do not become too heavy during the
optimisation. More precisely, in Lemma 2 we translated the kurtosis condition to a condition that is more easily
verifiable in the case of exponential families and in Lemma 3 we explicitly proved that the condition is fulfilled in the
Gaussian case (where the 1-dimensional case carries over to the multidimensional case). In practice, $q$ is often modelled
by a Gaussian (potentially with a neural network outputting its mean and covariance). The analysis for the Gaussian
case can be found in Section C. Note that in higher dimensions (usually implying that $\mathrm{KL}(q\|p)$ is large, as elaborated
in Lemma 4) the kurtosis is allowed to be large (but must be bounded—see also Corollary 1). More importantly, our
numerical experiments suggest that the kurtosis condition is satisfied for more complicated cases (see Figures 1, 2).

*R1: The order notation might hide large constants; elaborate the significance of Corollary 1.* Numerical evidence
suggests that, in practice, the ratio $\delta^{\mathrm{CV}}/\mathbb{E}[a_{\mathrm{VarGrad}}]$ is usually very small (e.g., Figure B.4 shows that the order of
magnitude is $10^{-8}$, and we have observed similar results in all other experiments). Corollary 1 offers a rigorous
comparison of the variances of Reinforce and VarGrad that does not depend on assumptions that might be hard to check.

*R1, R2, R3, R4: Empirical evaluation.* Please see the included figure for an
empirical validation of $\delta^{\mathrm{CV}}/\mathbb{E}[a_{\mathrm{VarGrad}}]$ on a non-linear (deep) DVAE model
used in Tucker, et. al. [2017] on Omniglot. In the attached plot this quantity
is typically very small and fluctuates around zero, validating our analytical
results in the challenging non-linear setting (a similar result is shown in Figure
B.4 for the two-layer linear case with an in-depth discussion that follows for
the non-linear case too). For the non-linear case, we also observe practical
performance gains similar to the linear case in the trace plots. The new results
have now been added to the manuscript.

*R1: Is the log-variance well-defined?* Since $p$, $q$ and $r$ are assumed to admit densities, it follows (under the support
condition stated in the paper) that measure-zero sets of $r$ are necessarily measure-zero sets of $p$ and $q$, implying the
divergence property. This has now been clarified in the paper.

*R3: VarGrad does not account for model structure.* The advantage of VarGrad is that it is easily implementable and
applicable to a large array of models (black-box). In Figures 3, B.7 and B.8 we demonstrate that it is still competitive
w.r.t. to alternative (tuned and structured) estimators.

*R3: Deeper analysis w.r.t. the choice of $r$.* We agree; however, as our work was focused on the theoretical analysis of
the induced ELBO gradient estimator rather than a new variational objective, we left this question out intentionally for
subsequent studies, in order to not confuse the presentation. Taking $r = q$ after differentiation reproduces the gradient
of the KL divergence/ELBO (Proposition 1) and this choice was shown to be effective in the empirical evaluation
(Nüsken & Richter [2020] provide further arguments for this choice).

*R4: Comparison to reparametrisation.* The reparametrisation trick is often not applicable (e.g., in discrete models),
whereas VarGrad is general-purpose. That said, in Figures 3, B.7 and B.8 we compared against RELAX + REBAR,
which use reparametrisation estimators as control variates.

*R4: Application to mixtures of non-diagonal Gaussians.* Indeed this is a potential application, since VarGrad can be
applied easily to many variational families including non-diagonal mixtures of Gaussians. We will explore this further.

*R1: Figure 1 error bars and clarification of the oracle estimator.* We have now added the error bars. The oracle
estimator in Figure 2 takes 1000 *extra* samples. We have now clarified this in the figure caption.

*R1: Why did you focus on the ELBO in the presentation?* As per your suggestion, we have now shifted the presentation
on the ELBO to streamline the background section. Extra details were moved to the appendix.

[Meta-Review · NeurIPS 2020]

This paper presents a novel analysis of an existing algorithm, showing that it can be reinterpreted. In particular, take the leave-one-out control variate to reduce the variance of the score-function estimator. This can be seen as a "naive" gradient of a new log-variance loss. This appears to be a novel and valuable contribution.